# Metallic ferromagnetic films with magnetic damping under $1.4 \times 10^{-3}$

Aidan J. Lee[1], Jack T. Brangham[1], Yang Cheng[1], Shane P. White[1], William T. Ruane[1], Bryan D. Esser [2], David W. McComb[2], P. Chris Hammel[1] & Fengyuan Yang [1]

Low-damping magnetic materials have been widely used in microwave and spintronic applications because of their low energy loss and high sensitivity. While the Gilbert damping constant can reach $10^{-4}$ to $10^{-5}$ in some insulating ferromagnets, metallic ferromagnets generally have larger damping due to magnon scattering by conduction electrons. Meanwhile, low-damping metallic ferromagnets are desired for charge-based spintronic devices. Here, we report the growth of $Co_{25}Fe_{75}$ epitaxial films with excellent crystalline quality evident by the clear Laue oscillations and exceptionally narrow rocking curve in the X-ray diffraction scans as well as from scanning transmission electron microscopy. Remarkably, the $Co_{25}Fe_{75}$ epitaxial films exhibit a damping constant $<1.4 \times 10^{-3}$, which is comparable to the values for some high-quality $Y_3Fe_5O_{12}$ films. This record low damping for metallic ferromagnets offers new opportunities for charge-based applications such as spin-transfer-torque-induced switching and magnetic oscillations.

[1] Department of Physics, The Ohio State University, Columbus, OH 43210, USA. [2] Center for Electron Microscopy and Analysis, Department of Materials Science and Engineering, The Ohio State University, Columbus, OH 43210, USA. Aidan J. Lee and Jack T. Brangham contributed equally to this work. Correspondence and requests for materials should be addressed to F.Y. (email: yang.1006@osu.edu)

nsulating ferromagnets (FMs) with low Gilbert damping, such as $Y_3Fe_5O_{12}$ (YIG), are advantageous for certain applications[1-11]; however, they are not suitable for spintronic devices based on charge currents, which require metallic FMs[12-17]. Ultralow-damping metallic FMs are desirable for spin-transfer-torque-induced magnetic switching and dynamics in magnetic multilayers[18] and FM/heavy-metal structures[19] because of the lower critical current needed. Previously, Gilbert damping constants ($\alpha$) as low as $1.9 \times 10^{-3}$ have been reported for epitaxial films of Fe and Fe alloys[20-22]. Recently, Schoen et al.[23] studied magnetic damping in polycrystalline $Co_xFe_{1-x}$ films of various compositions and measured a minimum $\alpha = 2.1 \times 10^{-3}$ for $Co_{25}Fe_{75}$[24-26]. This ultralow damping is attributed to a minimum in the electronic density of states at the Fermi energy for $x = 25\%$, since the intrinsic Gilbert damping has been shown to be proportional to the density of states at the Fermi energy in $Co_xFe_{1-x}$.

These results from polycrystalline films motivated the studies reported here that were guided by the expectation that the lower defect density in high-quality epitaxial films of $Co_{25}Fe_{75}$ will lead to reduced damping[22] as compared to their polycrystalline counterparts. In this study we grow $Co_{25}Fe_{75}$ epitaxial films using off-axis sputtering on two kinds of substrates with the goal of further reducing the magnetic damping of this promising metallic FM towards an unprecedented level. X-ray diffraction (XRD) and scanning transmission electron microscopy (STEM) verify that these films are single crystal with high crystalline quality. Variable frequency FM resonance (FMR) measurements confirm that these films exhibit significantly reduced Gilbert damping—$< 1.4 \times 10^{-3}$—in contrast to polycrystalline films.

## Results

**Epitaxial growth of $Co_{25}Fe_{75}$.** The growths were done using ultrahigh vacuum, off-axis sputtering with a base pressure lower than $2 \times 10^{-10}$ Torr, which has previously been used to grow high-quality epitaxial films of various metals and oxides[27-32]. A $Co_{25}Fe_{75}$ sputter target 5 cm in diameter was prepared by annealing a pressed target of a stoichiometric Co and Fe powder mixture at 600 °C under $H_2$ gas flow. The $Co_{25}Fe_{75}$ target was mounted on a horizontal sputtering source. The substrate was positioned at a horizontal distance of 5.4 cm from the target and 7.5 cm below the central axis of the target with the sample surface perpendicular to the target surface, which results in an average deposition angle of 54° relative to the target normal. We first grew 10 nm polycrystalline $Co_{25}Fe_{75}$ films on Si at room temperature with a 3 nm Cu seed layer and 5 nm Cr cap, which exhibited a low damping constant similar to that previously reported[23]. The Cr, Cu, and $Co_{25}Fe_{75}$ films were grown at Ar pressures of 10, 5, and 10 mTorr using DC sputtering with growth rates of 1.7, 3.6, and 2.4 nm/min, respectively. $Co_{25}Fe_{75}$ epitaxial films of 6.8 and 34 nm thicknesses were deposited at a substrate temperature of 300 °C directly on (001)-oriented MgO and $MgAl_2O_4$ (MAO) substrates (purchased from MTI), followed by a Cr capping layer grown at room temperature. MgO is a commonly used substrate with a lattice constant of $a = 4.212$ Å, which has a 3.9% lattice mismatch with $Co_{25}Fe_{75}$ when considering the 45° rotation between the body-centered-cubic (BCC) lattice of $Co_{25}Fe_{75}$ and the face-centered-cubic (FCC) lattice of MgO. $MgAl_2O_4$ ($a = 8.083$ Å) was used because it has a much better lattice match (within 0.4%) with $Co_{25}Fe_{75}$.

**Characterization of crystalline quality.** The crystalline quality of the epitaxial $Co_{25}Fe_{75}$ films was quantified by several methods. Figure 1a shows $2\theta-\omega$ scans obtained using triple-axis X-ray diffractometry of $Cr(2.8 \text{ nm})/Co_{25}Fe_{75}(34 \text{ nm})$ and

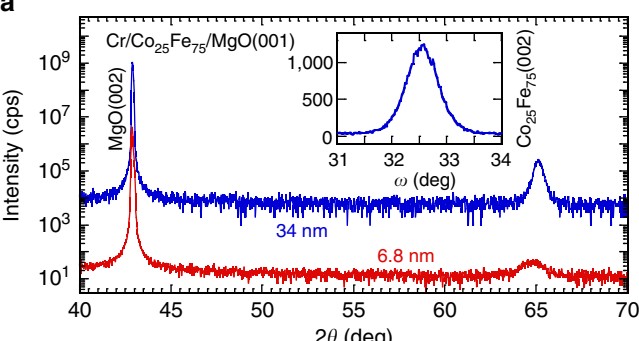

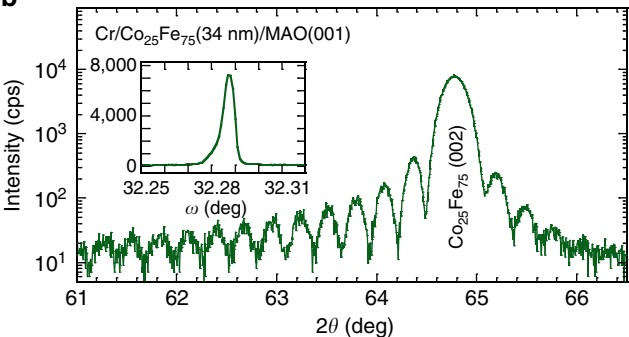

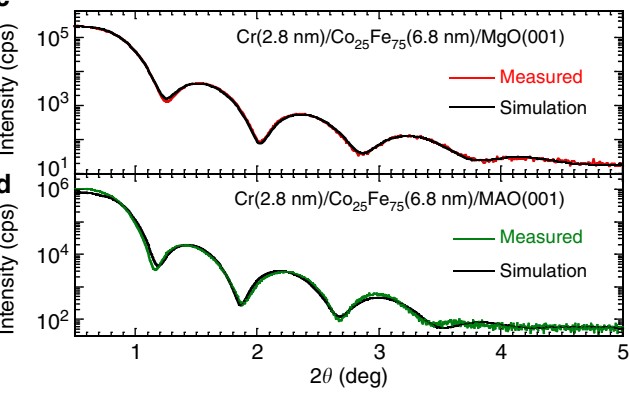

**Fig. 1** X-ray diffractometry and reflectometry of $Co_{25}Fe_{75}$ films on MgO and $MgAl_2O_4$ (MAO). **a** $2\theta-\omega$ X-ray diffraction (XRD) scans of $Cr(2.8 \text{ nm})/Co_{25}Fe_{75}/MgO(001)$ films with the $Co_{25}Fe_{75}$ thickness of 6.8 and 34 nm. Inset: XRD rocking curve of the $Co_{25}Fe_{75}(002)$ peak of the 34 nm film gives a full-width-half-maximum (FWHM) of 0.68°. **b** $2\theta-\omega$ scan of a $Cr(2.8 \text{ nm})/Co_{25}Fe_{75}(34 \text{ nm})/MgAl_2O_4(001)$ sample. Inset: XRD rocking curve of the $Co_{25}Fe_{75}(002)$ peak gives a FWHM of 0.0057°. X-ray reflectometry scans of **c** $Cr(2.8 \text{ nm})/Co_{25}Fe_{75}(6.8 \text{ nm})/MgO(001)$ and **d** $Cr(2.8 \text{ nm})/Co_{25}Fe_{75}(6.8 \text{ nm})/MAO(001)$ films, where a corresponding fit (black) gives a surface roughness of 0.4 nm for both samples

$Cr(2.8 \text{ nm})/Co_{25}Fe_{75}(6.8 \text{ nm})$ films on MgO(001). The $Co_{25}Fe_{75}(002)$ peaks at 65.139° and 64.844° for the 34 and 6.8 nm films correspond to out-of-plane BCC lattice constants of 2.862 and 2.873 Å, respectively. The XRD rocking curve (inset to Fig. 1a) of the $Cr(2.8 \text{ nm})/Co_{25}Fe_{75}(34 \text{ nm})/MgO$ sample gives a full-width-at-half-maximum (FWHM) of 0.68°. Figure 1b shows a $2\theta-\omega$ scan of a $Cr(2.8 \text{ nm})/Co_{25}Fe_{75}(34 \text{ nm})/MAO(001)$ sample, which exhibits clear Laue oscillations and a rocking curve FWHM of only 0.0057° (inset to Fig. 1b). The pronounced Laue oscillations and exceptionally narrow rocking curve are rarely seen in metallic epitaxial films, demonstrating very high crystalline quality enabled by the excellent lattice match between $Co_{25}Fe_{75}$ and MAO. Figure 1c, d show the X-ray reflectometry

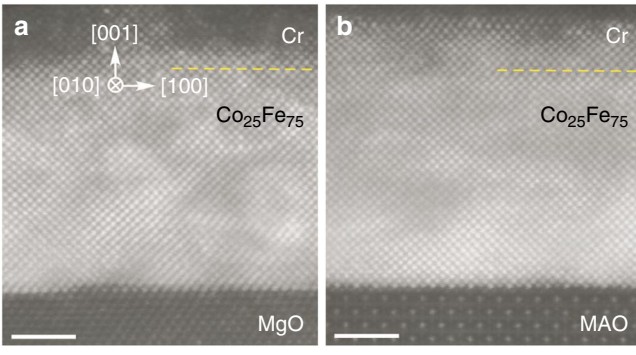

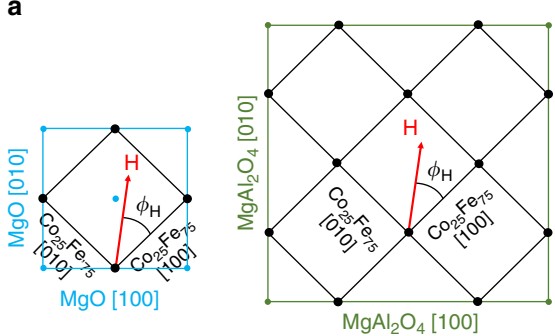

**Fig. 2** Atomic resolution images of $Co_{25}Fe_{75}$ films. Scanning transmission electron microscopy images of **a** Cr(2.8 nm)/$Co_{25}Fe_{75}$(6.8 nm)/MgO and **b** Cr(2.8 nm)/$Co_{25}Fe_{75}$(6.8 nm)/MAO films viewed along the $Co_{25}Fe_{75}$ [010] axis, where the boundary between Cr and $Co_{25}Fe_{75}$ as confirmed by electron energy loss spectroscopy scans is indicated by the dashed line. In **a**, the less clear $Co_{25}Fe_{75}$ atomic columns near the middle within 2 nm from the interface is due to strain relaxation via the disappearance of a $Co_{25}Fe_{75}$ column (MgO lattice constant is 3.9% larger than that of $Co_{25}Fe_{75}$). In **b**, the $Co_{25}Fe_{75}$ lattice matches perfectly with that of MAO without strain relaxation and its associated defects. *Scale bars*: 2 nm

scans of a Cr(2.8 nm)/$Co_{25}Fe_{75}$(6.8 nm) bilayer on MgO and MAO, respectively, where the fitting gives a film roughness of 0.4 nm for both samples.

The Cr(2.8 nm)/$Co_{25}Fe_{75}$(6.8 nm) films on MgO and MAO were characterized by high-angle annular dark field scanning transmission electron microscopy (STEM) using an FEI probe-corrected Titan[3] 80–300 S/TEM. Figure 2 shows the STEM images viewed along the $Co_{25}Fe_{75}$ [010] direction (MgO and MAO [110] axis), which reveal the BCC ordering of $Co_{25}Fe_{75}$ and its epitaxy on the substrates. Electron energy loss spectroscopy was used to determine the Cr/$Co_{25}Fe_{75}$ boundary, which is unclear in the STEM images because of the small difference in their atomic numbers. The STEM images also show the epitaxial relationship between $Co_{25}Fe_{75}$ and the substrates as illustrated in Fig. 3a, where the BCC lattice of $Co_{25}Fe_{75}$ grows at a 45° in-plane rotation relative to the FCC lattice of MgO and the spinel lattice of MAO.

## Characterization of in-plane magnetization.

Magnetic hysteresis loops of Cr(2.8 nm)/$Co_{25}Fe_{75}$(6.8 nm) on MgO and MAO were measured at room temperature using a vibrating sample magnetometer at various in-plane orientations defined by $\phi_H$, the angle between the applied magnetic field ($H$) and the $Co_{25}Fe_{75}$ [100] axis (see Fig. 3a). Figure 3b shows two hysteresis loops for Cr(2.8 nm)/$Co_{25}Fe_{75}$(6.8 nm)/MgO measured at $\phi_H = 0°$ and 45° where the small diamagnetic background from MgO has been subtracted. The $\phi_H = 0°$ loop has a sharp magnetic reversal with a coercive field ($H_c$) of 20 Oe while the $\phi_H = 45°$ loop has a higher saturation field and $H_c = 23$ Oe. By comparing the two hysteresis loops, we conclude that $Co_{25}Fe_{75}$ [100] ($\phi_H = 0°$) is the easy axis and $Co_{25}Fe_{75}$ [110] ($\phi_H = 45°$) is the in-plane hard axis within the framework of in-plane cubic anisotropy. The saturation magnetization, $4\pi M_s = 2.46 \pm 0.02$ T, of the film on MgO is extracted from the hysteresis loop, which has been confirmed by a measurement using a SQUID magnetometer.

The inset to Fig. 3b shows a closer view of the hard-axis loop ($\phi_H = 45°$), which allows us to obtain the in-plane magnetocrystalline anisotropy. By applying separate linear fits to the saturated high field regime and the region between 200 and 0 Oe, we extract the in-plane magnetocrystalline anisotropy of $260 \pm 10$ Oe between the [100] and [110] axes[33]. This is in agreement with the

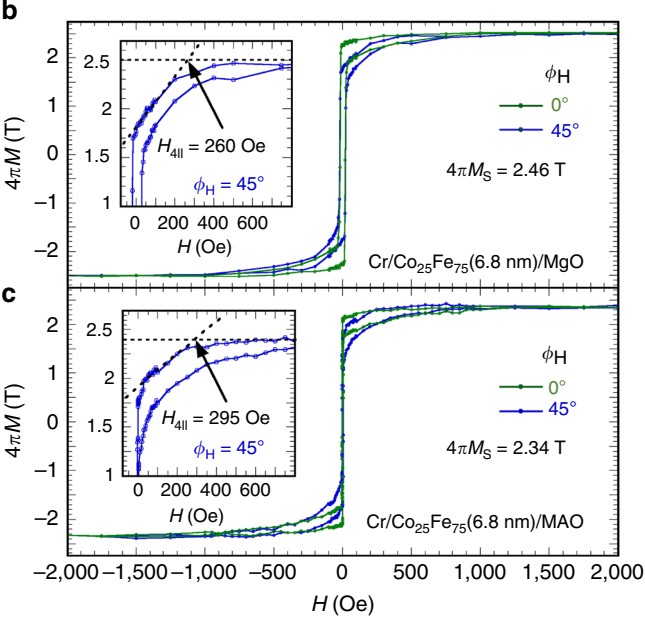

**Fig. 3** Determination of in-plane magnetic hard axis. **a** Schematic illustration of the epitaxial relationship between the $Co_{25}Fe_{75}$ lattice and the MgO and MAO(001) substrates as well as the in-plane field angle $\phi_H$ with respect to the $Co_{25}Fe_{75}$ [100] axis. In-plane magnetic hysteresis loops for the **b** Cr(2.8 nm)/$Co_{25}Fe_{75}$(6.8 nm)/MgO(001) and **c** Cr(2.8 nm)/$Co_{25}Fe_{75}$(6.8 nm)/MAO(001) sample for field applied along a $Co_{25}Fe_{75}$ [100] axis ($\phi_H = 0°$, easy axis) and a $Co_{25}Fe_{75}$ [110] axis ($\phi_H = 45°$, hard axis). A saturation magnetization of 2.46 and 2.34 T as well as an easy-axis coercive field of 20 and 2.5 Oe are obtained for the 6.8 nm $Co_{25}Fe_{75}$ film on MgO and MAO, respectively. The insets in **b** and **c** show a zoom-in view of the hard-axes hysteresis loops ($\phi_H = 45°$) to extract the in-plane magnetocrystalline anisotropy, $H_{4\parallel} = 260$ and 295 Oe, for the $Co_{25}Fe_{75}$ film on MgO and MAO, respectively

in-plane cubic anisotropy $H_{4\parallel}$ obtained from FMR measurements described below. Figure 3c shows the in-plane hysteresis loops for Cr(2.8 nm)/$Co_{25}Fe_{75}$(6.8 nm)/MAO, from which we obtain $4\pi M_s = 2.34 \pm 0.02$ T and an in-plane magnetocrystalline anisotropy of $295 \pm 10$ Oe.

To probe the dynamic magnetic properties of the $Co_{25}Fe_{75}$ epitaxial films, we performed angular-dependent FMR measurements on our films. Figure 4a shows the derivative spectra of FMR absorption for the Cr(2.8 nm)/$Co_{25}Fe_{75}$(6.8 nm)/MgO sample for various $\phi_H$ from −1° to 45° taken at a microwave frequency of $f = 9.66$ GHz in a cavity, from which the resonant field ($H_{res}$) is obtained. From the in-plane angular dependence of $H_{res}$, we can determine the magnetic anisotropy of the $Co_{25}Fe_{75}$ film. Magnetization subject to an energy landscape with cubic

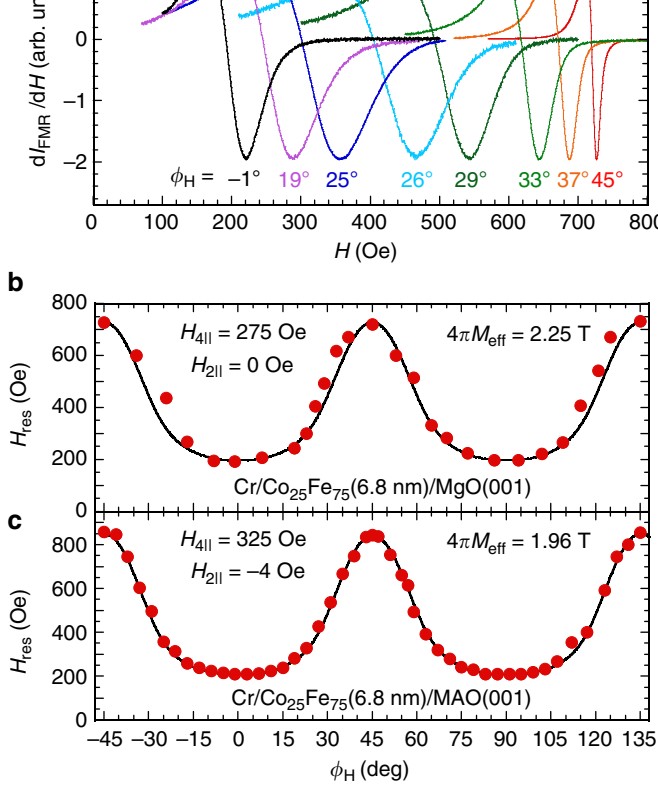

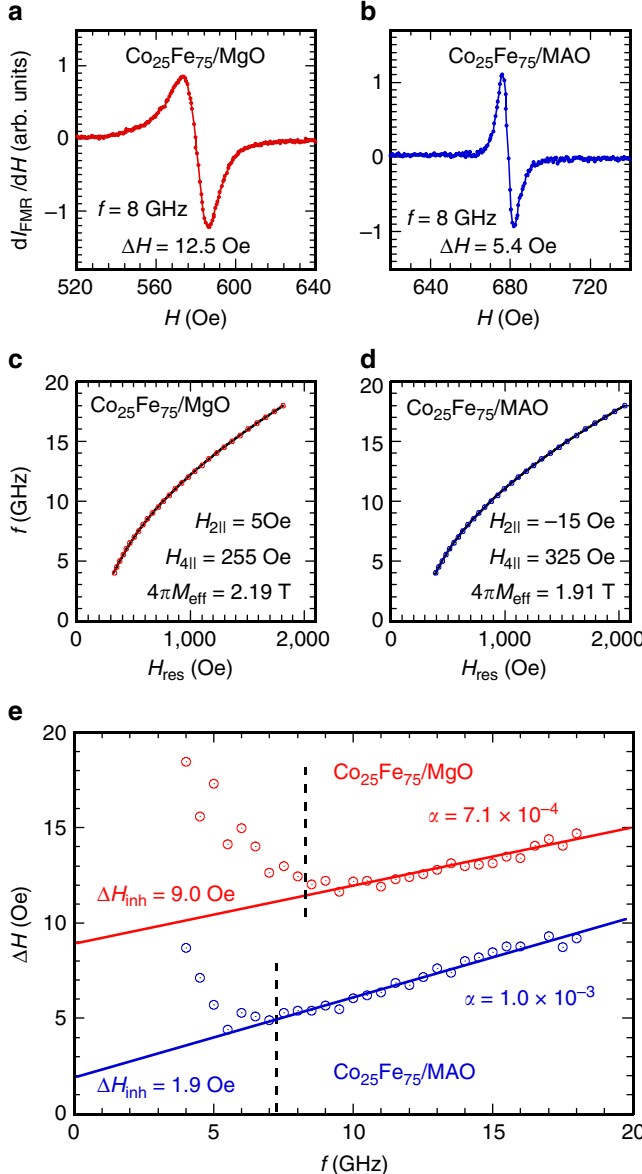

**Fig. 4** Angular dependence of ferromagnetic resonance. **a** Derivative ferromagnetic resonance (FMR) absorption spectra for a Cr(2.8 nm)/Co$_{25}$Fe$_{75}$(6.8 nm)/MgO(001) sample at in-plane angles $\phi_H$ from −1° to 45°. **b** The resonance fields of the FMR spectra in **a** plotted against $\phi_H$. From the fit to the experimental data the values for $H_{4\parallel}$, $H_{2\parallel}$, and $4\pi M_{eff}$ are extracted. **c** A similar $H_{res}$ vs. $\phi_H$ plot for a Cr(2.8 nm)/Co$_{25}$Fe$_{75}$(6.8 nm)/MAO(001) sample

symmetry can be quantitatively described by the total free energy density ($F$) given by[15, 34],

$$F = -\mathbf{H} \cdot \mathbf{M} + \frac{1}{2}M\left\{4\pi M_{eff}\cos^2\theta - \frac{1}{2}H_{4\perp}\cos^4\theta \right.$$

$$\left. - \frac{1}{8}H_{4\parallel}(3 + \cos^4\phi)\sin^4\theta - H_{2\parallel}\sin^2\theta\sin^2\left(\phi - \frac{\pi}{4}\right)\right\}, \quad (1)$$

where $\theta$ is the out-of-plane angle and $\phi$ the in-plane angle for the orientation of the equilibrium magnetization (**M**) with respect to the easy axis, $4\pi M_{eff} = 4\pi M_s - H_{2\perp}$ is the effective saturation magnetization, and $H_{2\perp}$ is the out-of-plane uniaxial anisotropy. $H_{4\perp}$, $H_{4\parallel}$, and $H_{2\parallel}$ are the out-of-plane cubic, in-plane cubic, and in-plane uniaxial anisotropy, respectively. By substituting Eq. (1) into the FMR resonance condition[15, 34],

$$\left(\frac{\omega}{\gamma}\right)^2 = \frac{1}{M^2\sin^2\theta}\left[\frac{\partial^2 F}{\partial\theta^2}\frac{\partial^2 F}{\partial\phi^2} - \left(\frac{\partial^2 F}{\partial\phi\partial\theta}\right)^2\right], \quad (2)$$

we can calculate $H_{res}$ for a given $\phi_H$, where $\gamma$ is the gyromagnetic ratio and $\omega$ is the resonance angular frequency. For in-plane angular FMR, $\theta = 90°$ and the $H_{4\perp}$ term drops out; the in-plane equilibrium angle $\phi$ is determined by numerically minimizing the free energy density.

Figure 4b shows the in-plane angular dependence of $H_{res}$ for Cr(2.8 nm)/Co$_{25}$Fe$_{75}$(6.8 nm)/MgO and the fit using Eq. (2) and

**Fig. 5** Determination of Gilbert damping. Representative ferromagnetic resonance (FMR) absorption derivative spectra for the **a** Cr(2.8 nm)/Co$_{25}$Fe$_{75}$(6.8 nm)/MgO(001) and **b** Cr(2.8 nm)/Co$_{25}$Fe$_{75}$(6.8 nm)/MAO (001) sample at $\phi_H = 45°$ measured at $f = 8$ GHz, which give a FMR linewidth of 12.5 and 5.4 Oe, respectively. Frequency vs. resonance field plots for the Co$_{25}$Fe$_{75}$ films on **c** MgO and **d** MAO measured at $\phi_H = 45°$, from which the anisotropy terms and effective saturation magnetization are extracted. **e** Frequency dependencies of the FMR linewidths for the Co$_{25}$Fe$_{75}$ films on MgO (red) and MAO (blue), where the behavior below 8 and 7 GHz, respectively, reflects incomplete saturation of the films leading to increases in the inhomogeneous broadening. The solid lines are linear fits to the frequency range above 8 (7) GHz for the film on MgO (MAO), from which the Gilbert damping and inhomogeneous broadening are extracted

$\gamma/2\pi = 29.5 \pm 1.0$ GHz T$^{-1}$[35], which agrees well with the experimental data and gives $H_{2\parallel} = 0 \pm 5$ Oe, $H_{4\parallel} = 275 \pm 5$ Oe, and $4\pi M_{eff} = 2.25 \pm 0.02$ T. The obtained $H_{4\parallel}$ agrees with the value (260 Oe) determined from the magnetometry measurements in Fig 3b. Figure 4c shows the in-plane angular dependence of $H_{res}$ for Cr(2.8 nm)/Co$_{25}$Fe$_{75}$(6.8 nm)/MAO. From the fitting to the data, we obtain $H_{2\parallel} = -5 \pm 5$ Oe, $H_{4\parallel} = 325 \pm 5$ Oe, and $4\pi M_{eff} = 1.96 \pm 0.02$ T, where the value of $H_{4\parallel}$ is close to the anisotropy (295 Oe) obtained in Fig. 3c.

**Measurement of Gilbert damping.** Frequency-dependent FMR absorption was measured between 3 and 18 GHz using a microwave stripline by sweeping the magnetic field at various fixed frequencies. A small modulation of the magnetic field was applied to enable the measurement of differential absorbed power with a Schottky diode detector and a lock-in amplifier. Figure 5a, b show representative FMR spectra at $f = 8$ GHz with the magnetic field applied along the $Co_{25}Fe_{75}$ [110] axis for the $Cr(2.8 \text{ nm})/Co_{25}Fe_{75}(6.8 \text{ nm})/MgO$ and $Cr(2.8 \text{ nm})/Co_{25}Fe_{75}(6.8 \text{ nm})/MAO$ samples, which exhibit a peak-to-peak linewidth ($\Delta H$) of 12.5 and 5.4 Oe, respectively. The FMR linewidth of 5.4 Oe at 8 GHz rivals that of high-quality YIG films and is unprecedented for metallic FM films. We fit the $f$ vs. $H_{res}$ plots in Fig. 5c, d using the same procedure for Fig. 4. The extracted values for in-plane anisotropies and effective saturation magnetization are: $H_{4\parallel} = 260 \pm 5$ Oe, $H_{2\parallel} = 0 \pm 5$ Oe, and $4\pi M_{eff} = 2.19 \pm 0.02$ T for the film on MgO; $H_{4\parallel} = 315 \pm 10$ Oe, $H_{2\parallel} = -5 \pm 5$ Oe, and $4\pi M_{eff} = 1.91 \pm 0.02$ T for the film on MAO, which agree with the results in Fig. 4.

Figure 5e shows the corresponding frequency dependencies of $\Delta H$ for the two samples. Below ~8 (7) GHz for the film on MgO (MAO), the data deviate from the linear dependence found above it. When comparing the hard-axis saturation fields of ~ 600 Oe to the $H_{res}$ of 581 Oe (592 Oe) at 8 (7) GHz for the film on MgO (MAO), we can understand these two regimes as follows. The films exhibit linear frequency dependence of $\Delta H$ above a threshold field where the magnetization is fully aligned. Below the threshold, the magnetization is not fully aligned, leading to non-uniform magnetization which inhomogeneously broadens the linewidth[16]. We first apply a linear fit in Fig. 5e in the frequency regime above 8 (7) GHz for the film on MgO (MAO), from which the damping constant can be determined using $\Delta H = \Delta H_{inh} + \frac{4\pi\alpha f}{\sqrt{3}\gamma}$, where $\Delta H_{inh}$ is the inhomogeneous broadening[36]. For $Co_{25}Fe_{75}(6.8 \text{ nm})/MgO$, $\alpha = (7.1 \pm 0.6) \times 10^{-4}$ and $\Delta H_{inh} = 9.2 \pm 0.3$ Oe; for $Co_{25}Fe_{75}(6.8 \text{ nm})/MAO$, $\alpha = (1.16 \pm 0.02) \times 10^{-3}$ and $\Delta H_{inh} = 1.8 \pm 0.1$ Oe. If we choose a higher threshold field corresponding to 10 (9) GHz for the film on MgO (MAO), we obtain $\alpha = (8.5 \pm 0.6) \times 10^{-4}$ and $\Delta H_{inh} = 8.4 \pm 0.3$ Oe for the film on MgO, and $\alpha = (1.21 \pm 0.02) \times 10^{-3}$ and $\Delta H_{inh} = 1.5 \pm 0.1$ Oe for the film on MAO. The small $\Delta H_{inh}$ in the $Co_{25}Fe_{75}$ film on MAO can be attributed to the high crystalline quality of the film. The damping constant in the film on MAO is slightly larger than that on MgO, for which the reason is unknown at this point. We note that FMR measurements at higher frequencies, e.g., 40 or 70 GHz, would further improve the accuracy for the extracted values of intrinsic damping in our films; however, we do not have this capability and will pursue such measurements through collaboration in the future. Based on our measurements, we are confident that the intrinsic damping constant in our epitaxial $Co_{25}Fe_{75}$ films is below $1.4 \times 10^{-3}$, which is obtained when we assume $\Delta H_{inh} = 0$ for $Co_{25}Fe_{75}/MAO$.

## Discussion

The measured Gilbert damping constant—$<1 \times 10^{-3}$—is extremely low for metallic FMs and, remarkably, is comparable to those reported for YIG films[7]. The lowest $\alpha$ ($7.1 \times 10^{-4}$) for our epitaxial $Co_{25}Fe_{75}$ films is considerably lower than the values reported for polycrystalline $Co_{25}Fe_{75}$ films[23] ($2.1 \times 10^{-3}$) and epitaxial Fe films ($1.9 \times 10^{-3}$)[22]. Considering our measured damping constant also includes contributions from spin pumping into the Cr capping layer[37] and radiative loss into the environment and stripline[23], the intrinsic damping of our $Co_{25}Fe_{75}$ films should approach the calculated intrinsic value in the mid-low $10^{-4}$ regime[23–26].

In summary, we observe extremely low magnetic damping in $Co_{25}Fe_{75}$ epitaxial thin films with excellent crystalline quality.

This is the first direct measurement of a Gilbert damping constant in the $10^{-4}$ regime for metallic FM films, making it an ideal material for exploration of spintronic applications requiring metallic FMs.

**Data availability.** Data that support the findings of this report are available from the corresponding author upon request.

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

## Acknowledgements

This work was supported by National Science Foundation under Grant No. DMR-1507274 (sample growth and characterization, magnetization measurements, and data analysis) and by US Department of Energy, Office of Science, Basic Energy Sciences, under Award No. DE-FG02–03ER46054 (FMR measurements). Partial support for this work was provided by the Center for Emergent Materials, an NSF-funded MRSEC, under Grant No. DMR-1420451 (STEM characterization).

## Author contributions

P.C.H. and F.Y. designed this study; A.J.L., J.T.B., and Y.C. prepared the samples and performed vibrating sample magnetometer, XRD, and angular FMR measurements; B.D.E. and D.W.M. contributed the STEM images; S.P.W. and W.T.R. preformed the frequency-dependent FMR measurements. All authors wrote the manuscript.

## Additional information

**Competing interests:** The authors declare no competing financial interests.

