## [Peer Review File · Nature Communications]

Reviewers' Comments:

Reviewer #1 (Remarks to the Author):

The manuscript by Yang and coworkers presents ferromagnetic resonance (FMR) measurements of relaxation in epitaxial Fe₇₅Co₂₅ films on two different substrates, MgO and MgAl₂O₄ (MAO.) The peak-to-peak FMR linewidths ΔH_{pp} are remarkably low in these films. I find the manuscript exciting, and wish to see it published in Nature Communications, but I recommend that the authors first moderate their claims a bit or present further measurements to justify them (authors' choice.) A second issue is that the work is not placed in proper context with prior studies of low-damping epitaxial Fe-alloys, including some rather similar work 20 years ago. Most likely this will imply minor revisions to facilitate rapid publication in Nature Communications.

In the manuscript, the authors measure a minimum linewidth to frequency ratio $\Delta H_{pp}/(\omega/2\pi)$ of 0.55 Oe/GHz for the film on MAO at 18 GHz (10 G pp linewidth). If all broadening mechanisms in the film are intrinsic at this frequency, i.e. attributed to a Gilbert α , the effective Gilbert $\alpha_{eff}=0.0014$, an upper bound for the intrinsic value. This is significantly lower than yet observed in epitaxial Fe, which has demonstrated both the lowest α_{eff} and the lowest fitted Gilbert α of any metallic ferromagnet. To my knowledge the minimum value of α_{eff} observed in epitaxial Fe is 41 G at 34 GHz; see Scheck et al APL 88 252510 (2006), for $\alpha_{eff}=0.003$. The intrinsic value demonstrated in Scheck et al PRL 98 117601 (2007) and prior references discussed therein is $\alpha=0.0019$. These papers showed that low damping could be realized in epitaxial metal films deposited by sputtering, rather than MBE, and provide some context for the present manuscript (which makes much the same point, but surpasses the prior work.)

I see the manuscript as a significant advance over prior work for two reasons:

1) It reduces attained values of α_{eff} in metallic ferromagnets by a factor of two, as described above.

2) It shows that Fe₇₅Co₂₅ alloys must indeed have damping lower than that of pure iron, something which was only suggested in the paper by Schoen, Nature Physics 2016 (ref. 18). The value actually measured there, $\alpha=0.0021$ up to 40 GHz, was higher than the value $\alpha=0.0018$ (G = 57 MHz) already measured in epitaxial Fe (and so not itself remarkable or

'ultralow'.) Low intrinsic values were found only after subtracting out other possible sources of intrinsic damping, a method of debatable validity. Yang et al show low values before subtracting any linewidths away.

However, the headline claim made by the authors, "Metallic Ferromagnetic Films with Magnetic Damping of 10^{-4} " requires further justification than presented in the manuscript. The authors should moderate this claim or present further measurements if they would like to retain it.

1) Even by the authors' own estimate of $\alpha=0.0007$, the damping is about an order of magnitude higher than what one might infer from the title (0.0001). "Damping of 10^{-4} " is unnecessarily confusing if what is meant is damping <0.001 (itself exciting enough.)

2) The frequency range studied here (to 18 GHz) is too limited for the authors to be able to identify the intrinsic Gilbert α with high confidence. In studies of low-damping epitaxial Fe, measurements up to 70 GHz or even 90 GHz have been required in order to access a frequency range where the intrinsic linewidth (proportional to frequency) dominates over other mechanisms. In the measurements presented, this is not the case -- the Gilbert contribution appears to be a small fraction of the total linewidth (which is of course consistent with low values.)

High-frequency measurements are particularly necessary because of prior work. The Gilbert damping of epitaxial $\text{MgO}/\text{Fe}_{1-x}\text{Co}_x(20\text{nm})/\text{Au}$ films has already been studied across the whole composition, including $x=0.25$, up to 92 GHz, by F. Schreiber et al *Solid State Communications* 93 (12) p 965 1995. These films are nearly identical to those studied in the present manuscript except for Yang et al's substitution of the Cr cap. When the authors measure up to these very high frequencies, along the hard [110] axis, the compositional dependence of α (presented there as relaxation rate $G=\gamma M_s \alpha$) is featureless up to $x=0.5$, with minimum value 0.0018, i.e. that of pure Fe. They do not observe any minimum near Fe_3Co . Of course it may be that the epitaxial deposition in the present manuscript is superior, leading in some way to lower values of intrinsic damping (which seems contradictory to the meaning of 'intrinsic'), but it is difficult to be sure because the measurements presented in the manuscript are somewhat incompatible with those in the prior work.

So I urge the authors to perform measurements up to 70 GHz, or 90 GHz, or even above, to remove any remaining ambiguity.

Since the authors may not have access to these frequencies in the near term, I can also recommend publication of the paper with some minor revisions. If the authors claim only that they have measured a Gilbert damping <0.0014 , this is both well-justified by the present measurements and significant enough for the broad readership of *Nature Communications*.

Secondly the manuscript should be placed in context with prior work on epitaxial sputtered metal films with low damping, as discussed here.

Reviewer #2 (Remarks to the Author):

The manuscript reports growth of epitaxial films of Co₂₅Fe₇₅ with low Gilbert damping. This is a follow-up work on the recent report by another group (Ref. 18) showing that the damping in polycrystalline films of Co₂₅Fe₇₅ is low. The authors of the present manuscript demonstrate that the values reported in Ref. 18 can be further improved by a factor 2-3 if epitaxial Co₂₅Fe₇₅ films are used instead of polycrystalline films. This is a practically important result for the field of spintronics, although the material used is not new and the physics leading to the low intrinsic damping in this material was already explained in Ref. 18. Nevertheless, given the practical importance of this result, I can recommend this manuscript for publication if the authors properly address several issues discussed below. In my opinion, significant revisions of the manuscript are needed before it can be published.

1. I think the term “damping in the 10^{-4} regime” used by the authors in the title, abstract in discussion is misleading. The lowest damping they measured is much closer to 10^{-3} than to 10^{-4} . This unfortunate choice of language should be corrected.
2. Thickness of the Cr cap layer should be specified.
3. Are there contributions to the damping due to spin pumping into the Cr capping layer? What happens if an insulating capping layer is used? The authors should discuss the physics that leads to the measured values of the damping in detail. For example, Ref. 18 points out that the measured damping in their films is dominated by the spin pumping and the radiative damping contributions. What are the physical reasons the damping in the epitaxial films presented here is much lower in comparison to the polycrystalline films of Ref. 18? Is it due to reduction of the spin pumping or due to the epitaxial nature of the film? Some quantitative analysis of various contributions to the damping should be presented. Measuring films with different Co₂₅Fe₇₅ thicknesses and different capping layers can help in separation of various contributions to the damping.
4. The main result of the paper is growth of high quality epitaxial Co₂₅Fe₇₅ films with very low Gilbert damping by off-axis magnetron sputtering. However, few details of the sputter deposition method are given. The authors should add supplemental information showing the schematic of their deposition apparatus with the substrate-target distance, deposition angles, etc indicated in

the drawing.

5. The key data of this manuscript showing the FMR line width versus frequency are presented in Fig. 5e. The authors chose the applied field direction along the in-plane magnetically hard axis of the film ($\phi = 45$ degrees). This choice of the angle leads to incomplete saturation of magnetization at the low field/frequency values. The authors claim that this incomplete saturation is the reason for the observed deviations from linearity in the line width versus frequency. What motivates this choice of the applied field direction? The authors should also present data for the magnetic field applied along the easy axis of the magnetic anisotropy. It is unfortunate that the maximum frequency for these data is limited to 18 GHz, which leads to a very limited frequency range of the data used in the linear fitting. The authors should extend the frequency range to higher values.

6. The authors should also explain to the broad audience of the Nature Communications why metallic ferromagnets with low damping are of interest. A few examples in the introductory part of the manuscript would be very helpful.

Response to Reviews for NCOMMS-17-05391-T/Lee

We thank the two reviewers for their insightful reviews and encouraging comments about our manuscript, including “*I find the manuscript exciting, and wish to see it published in Nature Communications*” and “*Most likely this will imply minor revisions to facilitate rapid publication in Nature Communications*” from Reviewer 1 and “*...Given the practical importance of this result, I can recommend this manuscript for publication if the authors properly address several issues discussed below*” from Reviewer 2. Following the reviewers’ suggestions for improving the title, we have updated the phrasing in the title and throughout the manuscript to more accurately describe our findings as damping under 1×10^{-3} . Below, we address the reviewers’ questions and comments on a point-by-point basis.

Report of First Reviewer

- “*The manuscript by Yang and coworkers presents ferromagnetic resonance (FMR) measurements of relaxation in epitaxial Fe₇₅Co₂₅ films on two different substrates, MgO and MgAl₂O₄ (MAO.) The peak-to-peak FMR linewidths ΔH_{pp} are remarkably low in these films. I find the manuscript exciting, and wish to see it published in Nature Communications, but I recommend that the authors first moderate their claims a bit or present further measurements to justify them (authors' choice.)*”

Response: We thank the reviewer for their encouraging statements and have moderated the claims in the revised manuscript to more accurately state our findings.

- “*A second issue is that the work is not placed in proper context with prior studies of low-damping epitaxial Fe-alloys, including some rather similar work 20 years ago. Most likely this will imply minor revisions to facilitate rapid publication in Nature Communications. In the manuscript, the authors measure a minimum linewidth to frequency ratio $\Delta H_{pp}/(\omega/2\pi)$ of 0.55 Oe/GHz for the film on MAO at 18 GHz (10 G pp linewidth). If all broadening mechanisms in the film are intrinsic at this frequency, i.e. attributed to a Gilbert α , the effective Gilbert $\alpha_{eff}=0.0014$, an upper bound for the intrinsic value. This is significantly lower than yet observed in epitaxial Fe, which has demonstrated both the lowest α_{eff} and the lowest fitted Gilbert α of any metallic ferromagnet. To my knowledge the minimum value of α_{eff} observed in epitaxial Fe is 41 G at 34 GHz; see Scheck et al APL 88 252510 (2006), for $\alpha_{eff}=0.003$. The intrinsic value demonstrated in Scheck et al PRL 98 117601 (2007) and prior references discussed therein is $\alpha=0.0019$. These papers showed that low damping could be realized in epitaxial metal films deposited by sputtering, rather than MBE, and provide some context for the present manuscript (which makes much the same point, but surpasses the prior work.)*”

Response: As suggested by the reviewer, we have included three new references, Schreiber, *et al. Solid State Commun.* 93, 965 (1995), Scheck, *et al. Appl. Phys. Lett.* 88, 252510 (2006), and Scheck, *et al. Phys. Rev. Lett.* 98, 117601 (2007) and acknowledged the Gilbert damping constant as low as 1.9×10^{-3} reported in these papers in our revised introduction. Regarding the estimate for the upper bound of intrinsic damping, the reviewer’s recommendation works well for the film grown on MAO, which exhibits a small inhomogeneous broadening of only 1.8 Oe. However, for the film on MgO, the inhomogeneous broadening is 9.2 Oe; thus this simple method significantly overestimates the intrinsic value of damping. Therefore, we added near the end of the main text in the revised manuscript, “Based on our measurements, we are confident that the intrinsic

damping constant in our epitaxial Co₂₅Fe₇₅ films is below 1.4×10^{-3} , which is obtained when we assume $\Delta H_{\text{inh}} = 0$ for Co₂₅Fe₇₅/MAO.”

- *“I see the manuscript as a significant advance over prior work for two reasons:
1) It reduces attained values of α_{eff} in metallic ferromagnets by a factor of two, as described above.
2) It shows that Fe₇₅Co₂₅ alloys must indeed have damping lower than that of pure iron, something which was only suggested in the paper by Schoen, Nature Physics 2016 (ref. 18). The value actually measured there, $\alpha=0.0021$ up to 40 GHz, was higher than the value $\alpha=0.0018$ ($G = 57$ MHz) already measured in epitaxial Fe (and so not itself remarkable or 'ultralow'.) Low intrinsic values were found only after subtracting out other possible sources of intrinsic damping, a method of debatable validity. Yang et al show low values before subtracting any linewidths away.”*

Response: We thank the reviewer for recognizing the significance of this work.

- *“However, the headline claim made by the authors, "Metallic Ferromagnetic Films with Magnetic Damping of 10^{-4} " requires further justification than presented in the manuscript. The authors should moderate this claim or present further measurements if they would like to retain it.
1) Even by the authors' own estimate of $\alpha=0.0007$, the damping is about an order of magnitude higher than what one might infer from the title (0.0001). "Damping of 10^{-4} " is unnecessarily confusing if what is meant is damping <0.001 (itself exciting enough).”*

Response: We agree with the reviewer’s comment and have revised the title to “Metallic Ferromagnetic Films with Magnetic Damping Under 1×10^{-3} ” to more clearly represent the results that are, as reviewer 1 states in reference to this claim, “itself exciting enough”.

- *“2) The frequency range studied here (to 18 GHz) is too limited for the authors to be able to identify the intrinsic Gilbert α with high confidence. In studies of low-damping epitaxial Fe, measurements up to 70 GHz or even 90 GHz have been required in order to access a frequency range where the intrinsic linewidth (proportional to frequency) dominates over other mechanisms. In the measurements presented, this is not the case -- the Gilbert contribution appears to be a small fraction of the total linewidth (which is of course consistent with low values.)*

High-frequency measurements are particularly necessary because of prior work. The Gilbert damping of epitaxial MgO/Fe_{1-x}Co_x(20nm)/Au films has already been studied across the whole composition, including $x=0.25$, up to 92 GHz, by F. Schreiber et al Solid State Communications 93 (12) p 965 1995. These films are nearly identical to those studied in the present manuscript except for Yang et al's substitution of the Cr cap. When the authors measure up to these very high frequencies, along the hard [110] axis, the compositional dependence of α (presented there as relaxation rate $G=\gamma M_s \alpha$) is featureless up to $x=0.5$, with minimum value 0.0018, i.e. that of pure Fe. They do not observe any minimum near Fe₃Co. Of course it may be that the epitaxial deposition in the present manuscript is superior, leading in some way to lower values of intrinsic damping (which seems contradictory to the meaning of 'intrinsic'), but it is difficult to be sure because the measurements presented in the manuscript are somewhat incompatible with those in the prior work.

So I urge the authors to perform measurements up to 70 GHz, or 90 GHz, or even above, to remove any remaining ambiguity.

Since the authors may not have access to these frequencies in the near term, I can also recommend publication of the paper with some minor revisions. If the authors claim only that they have measured a Gilbert damping < 0.0014 , this is both well-justified by the present measurements and significant enough for the broad readership of Nature Communications. Secondly the manuscript should be placed in context with prior work on epitaxial sputtered metal films with low damping, as discussed here.”

Response: While we agree that higher frequency measurements would improve the confidence in the extracted value of intrinsic damping, we do not currently have the capability to measure FMR in that frequency regime. Fitting the ~10-18 GHz frequency range, while limited, is a reasonably accurate method of measuring the Gilbert damping and has often been used by researchers in the field. As suggested by the reviewer, we have added the following sentence near the end of the main text:

“We note that FMR measurements at higher frequencies, e.g., 40 or 70 GHz, would further improve the accuracy for the extracted values of intrinsic damping in our films; however, we do not have this capability and will pursue such measurements through collaboration in the future.”

Indeed, Schreiber, *et al.* [*Solid State Commun.* 93, 965 (1995)] studied the same materials system, $\text{Co}_x\text{Fe}_{1-x}$ epitaxial films as reported in our manuscript. However, we note two differences between their samples and ours: (1) their $\text{Co}_x\text{Fe}_{1-x}$ films are 20 nm thick and our $\text{Co}_{25}\text{Fe}_{75}$ films are 7 nm thick; we found that the $\text{Co}_{25}\text{Fe}_{75}$ films have the lowest damping around 7-nm thickness. (2) the FWHM of their XRD rocking curve is about 1° while our rocking curve FWHM is 0.68° for the film on MgO and 0.0057° for the film on MAO. In addition, the inhomogeneous broadening reported by Schreiber, *et al.* along the in-plane hard axis is over 100 Oe in their $\text{Fe}_{49}\text{Co}_{51}$ film on MgO, while our $\text{Co}_{25}\text{Fe}_{75}$ film on MgO has an in-plane hard axis inhomogeneous broadening of 9.2 Oe. These factors likely contribute to the different results between Schreiber, *et al.* and our manuscript.

Regarding the meaning of ‘intrinsic’, the quality of the samples, such as polycrystalline vs. epitaxial, crystalline uniformity, and composition homogeneity, does affect the Gilbert damping constant as well as the inhomogeneous broadening. We found that our epitaxial $\text{Co}_{25}\text{Fe}_{75}$ films have lower Gilbert damping than our polycrystalline $\text{Co}_{25}\text{Fe}_{75}$ films.

Following the reviewer’s suggestion, we added near the end of the main text in the revised manuscript, “Based on our measurements, we are confident that the intrinsic damping constant in our epitaxial $\text{Co}_{25}\text{Fe}_{75}$ films is below 1.4×10^{-3} , which is obtained when we assume $\Delta H_{\text{inh}} = 0$ for $\text{Co}_{25}\text{Fe}_{75}/\text{MAO}$.”

Report of Second Reviewer

- *“The manuscript reports growth of epitaxial films of $\text{Co}_{25}\text{Fe}_{75}$ with low Gilbert damping. This is a follow-up work on the recent report by another group (Ref. 18) showing that the damping in polycrystalline films of $\text{Co}_{25}\text{Fe}_{75}$ is low. The authors of the present manuscript demonstrate that the values reported in Ref. 18 can be further improved by a factor 2-3 if*

epitaxial Co₂₅Fe₇₅ films are used instead of polycrystalline films. This is a practically important result for the field of spintronics, although the material used is not new and the physics leading to the low intrinsic damping in this material was already explained in Ref. 18. Nevertheless, given the practical importance of this result, I can recommend this manuscript for publication if the authors properly address several issues discussed below. In my opinion, significant revisions of the manuscript are needed before it can be published.”

Response: We thank reviewer 2 for his/her endorsement of our manuscript and will address the concerns below.

- “1. I think the term “damping in the 10^{-4} regime” used by the authors in the title, abstract in discussion is misleading. The lowest damping they measured is much closer to 10^{-3} than to 10^{-4} . This unfortunate choice of language should be corrected. “

Response: This has been adjusted as requested. (Please see comments to reviewer 1 above).

- “2. Thickness of the Cr cap layer should be specified. “

Response: We followed the reviewer’s suggestion and added the Cr thickness in the main text and figure captions (not in the figures due to space constraint).

- “3. Are there contributions to the damping due to spin pumping into the Cr capping layer? What happens if an insulating capping layer is used? The authors should discuss the physics that leads to the measured values of the damping in detail. For example, Ref. 18 points out that the measured damping in their films is dominated by the spin pumping and the radiative damping contributions. What are the physical reasons the damping in the epitaxial films presented here is much lower in comparison to the polycrystalline films of Ref. 18? Is it due to reduction of the spin pumping or due to the epitaxial nature of the film? Some quantitative analysis of various contributions to the damping should be presented. Measuring films with different Co₂₅Fe₇₅ thicknesses and different capping layers can help in separation of various contributions to the damping.”

Response: This is an understandable concern as Ref. 18 does go into great detail about the sources of damping in an attempt to extract an “intrinsic” Gilbert damping value for comparison to theory. However, this comparison to theory is not the central focus of our manuscript. We wish to share the direct measurement of the ultralow Gilbert damping present in these films. In Ref. 18, they have a seed and capping layer, which both contain Cu and Ta. These layers will contribute a significant Gilbert damping enhancement due to spin pumping. In our work, we use a single 2.8 nm Cr capping layer (no need for a seed layer). From our previous work, Du, *et al. Phys. Rev. B* 90, 140407 (2014), we found that the magnetic damping enhancement of a YIG/Cr(10 nm) bilayer as compared to a single YIG layer is very small (see figure to the right). Thus, a 2.8 nm thick Cr capping layer should have an even smaller effect on the damping in the

Fig. 3(a) from *Phys. Rev. B* 90, 140407 (2014) shows that the damping of a YIG/Cr(10 nm) bilayer is very similar to that of a single YIG layer.

underlying $\text{Co}_{25}\text{Fe}_{75}$ film. Thus, a $\text{Co}_{25}\text{Fe}_{75}$ thickness dependence study for determining the spin pumping contribution of the Cr layer is not needed.

In addition, we found that the FMR linewidth of $\text{Co}_{25}\text{Fe}_{75}$ films shows a minimum at 7 nm thickness and increases considerably for thinner and thicker $\text{Co}_{25}\text{Fe}_{75}$ films, all with the same thickness of the Cr capping layer. Based on our results, we believe that the high crystalline quality of our films is the primary reason for the lower Gilbert damping compared to Ref. 18 due to a higher degree of homogeneity and lower density of defects.

- *“4. The main result of the paper is growth of high quality epitaxial $\text{Co}_{25}\text{Fe}_{75}$ films with very low Gilbert damping by off-axis magnetron sputtering. However, few details of the sputter deposition method are given. The authors should add supplemental information showing the schematic of their deposition apparatus with the substrate-target distance, deposition angles, etc indicated in the drawing.”*

Response: Following the suggestion by the reviewer, we have added in the bottom paragraph on page 2 the following sentences to describe the off-axis deposition geometry and other parameters:

“A $\text{Co}_{25}\text{Fe}_{75}$ sputter target 5 cm in diameter was prepared by annealing a pressed target of a stoichiometric Co and Fe powder mixture at 600°C under H_2 gas flow. The $\text{Co}_{25}\text{Fe}_{75}$ target was mounted on a horizontal sputtering source. The substrate was positioned at a horizontal distance of 5.4 cm from the target and 7.5 cm below the central axis of the target with the sample surface perpendicular to the target surface, which results in an average deposition angle of 54° relative to the target normal.”

This description gives all the details for the geometry of our film deposition. We would like to point out that we have a pending patent application regarding our off-axis sputtering technique, which includes the geometry of our chamber and growth conditions.

- *“5. The key data of this manuscript showing the FMR line width versus frequency are presented in Fig. 5e. The authors chose the applied field direction along the in-plane magnetically hard axis of the film ($\phi = 45$ degrees). This choice of the angle leads to incomplete saturation of magnetization at the low field/frequency values. The authors claim that this incomplete saturation is the reason for the observed deviations from linearity in the line width versus frequency. What motivates this choice of the applied field direction? The authors should also present data for the magnetic field applied along the easy axis of the magnetic anisotropy. It is unfortunate that the maximum frequency for these data is limited to 18 GHz, which leads to a very limited frequency range of the data used in the linear fitting. The authors should extend the frequency range to higher values. “*

Response: As can be seen in Figs. 4b and 4c, the resonant field for the easy axis ($\phi_H = 0^\circ$ and 90°) is about 200 Oe for both films at 9.66 GHz, while for the hard axis ($\phi_H = -45^\circ$, 45° and 135°), the resonant field is over 700 Oe. Although for an ideal film the magnetization would be saturated at very low field along the easy axis, in reality, it requires a 500-600 Oe field to saturate the whole film either along the easy or hard axis. This is likely due to the edges, corners, and defects. We did try FMR measurements along the easy axis, which gives resonant fields of less than 400 Oe for the frequency range up to 18 GHz, insufficient for full saturation. The FMR linewidths along the easy axis are typically large (> 30 Oe at 9.66 GHz) and show inconsistent (scattered) behavior. Since the hard-axis resonant field is the highest for a given frequency among all in-plane directions, we deliberately chose the hard axis which provides the largest frequency range within our

instrument capability for linear behavior in the linewidth vs. frequency plots. As shown in Figs. 5c-5e of our manuscript, above ~ 8 GHz (~ 600 Oe), the linewidth shows a linear dependence as a function of frequency. Also as mentioned above, we do not have the capability to do FMR measurements at higher frequencies and will pursue such measurements through collaboration in the future.

- *“6. The authors should also explain to the broad audience of the Nature Communications why metallic ferromagnets with low damping are of interest. A few examples in the introductory part of the manuscript would be very helpful.”*

Response: We have added the following sentence in the introduction with two additional references: “Ultralow damping metallic FMs are desirable for spin-transfer-torque induced magnetic switching and dynamics in magnetic multilayers¹⁹ and FM/heavy-metal structures²⁰ because of the lower critical current needed”.

Reviewers' Comments:

Reviewer #1 (Remarks to the Author):

I find the revisions to the main text after the abstract mostly satisfactory. The remaining issue with the manuscript as it stands is that the title and abstract have not been changed to be fully consistent with the findings reported in the main text.

In the revised main text, the authors concede that the lowest Gilbert damping that can be measured with confidence (in the MAO film) is 0.0014. Claiming "damping under 0.001" in the title, and alluding to 0.0007 in the abstract, does not represent the results well.

It may turn out that the damping for the epitaxial films is that low, or even lower, proven through high-frequency measurements up to more typical frequencies for Gilbert damping measurements of Fe (70 GHz). However, with the low-frequency data presented, it is hard to tell. (Rejoinder to the authors' rebuttal: 10-18 GHz is not a typical frequency range for damping measurements of Fe due to the small role of Gilbert damping and large role of inhomogeneous broadening; the situation is reversed for permalloy, and there 10-18 GHz is more typical and not as error-prone.)

To be explicit, "Metallic ferromagnetic films with magnetic damping under 1.4×10^{-3} " should be a perfectly acceptable title for Nature Communications -- it is noteworthy; there has never been a metallic film as good as this.

A more minor point: the sentence starting with "The recent discovery..." is a little confusing. Authors may want to distinguish between the discovery / report in [18] (of $\alpha \sim 0.0002$ for the alloy, which really is 'ultralow') and the direct measurement (of $\alpha \sim 0.0021$ for the heterostructure.)

In summary I recommend publication after these very minor revisions.

Reviewer #2 (Remarks to the Author):

The authors responded to most of the comments. It is unfortunate that they do not have capabilities of extending their FMR measurements to higher frequencies, which in my opinion places unnecessarily large error bars on the values of Gilbert damping obtained from their measurements performed in a very limited frequency range. Nevertheless I recommend the manuscript for publication in its revised form because it does demonstrate a record low FMR linewidth for a metallic ferromagnetic film that is not a half-metal. This paper should stimulate

further work in the field that will yield fully reliable values of the Gilbert damping via FMR measurements extended to the ~ 80-100 GHz frequency range.

Second Response to Reviews for NCOMMS-17-05391A

We thank the two reviewers again for their comments and suggestions. Below, we address the reviewers' comments on a point-by-point basis.

Report of First Reviewer

- *"I find the revisions to the main text after the abstract mostly satisfactory. The remaining issue with the manuscript as it stands is that the title and abstract have not been changed to be fully consistent with the findings reported in the main text.*

In the revised main text, the authors concede that the lowest Gilbert damping that can be measured with confidence (in the MAO film) is 0.0014. Claiming "damping under 0.001" in the title, and alluding to 0.0007 in the abstract, does not represent the results well.

It may turn out that the damping for the epitaxial films is that low, or even lower, proven through high-frequency measurements up to more typical frequencies for Gilbert damping measurements of Fe (70 GHz). However, with the low-frequency data presented, it is hard to tell. (Rejoinder to the authors' rebuttal: 10-18 GHz is not a typical frequency range for damping measurements of Fe due to the small role of Gilbert damping and large role of inhomogeneous broadening; the situation is reversed for permalloy, and there 10-18 GHz is more typical and not as error-prone.)

To be explicit, "Metallic ferromagnetic films with magnetic damping under 1.4×10^{-3} " should be a perfectly acceptable title for Nature Communications -- it is noteworthy; there has never been a metallic film as good as this."

Response: As suggested by the reviewer, we have changed the title to, "Metallic Ferromagnetic Films with Magnetic Damping Under 1.4×10^{-3} ". We have also modified our abstract, so it reads as the following, "Remarkably, the $\text{Co}_{25}\text{Fe}_{75}$ epitaxial films exhibit a damping constant less than 1.4×10^{-3} , which is comparable to the values for some high quality $\text{Y}_3\text{Fe}_5\text{O}_{12}$ films. This record low damping for metallic ferromagnets offers new opportunities for charge-based applications such as spin-transfer-torque induced switching and magnetic oscillations."

- *"A more minor point: the sentence starting with "The recent discovery..." is a little confusing. Authors may want to distinguish between the discovery / report in [18] (of $\alpha \sim 0.0002$ for the alloy, which really is 'ultralow') and the direct measurement (of $\alpha \sim 0.0021$ for the heterostructure.)"*

Response: We agree with the reviewer's comment and have removed the sentence from the abstract.

- *"In summary I recommend publication after these very minor revisions."*

Response: We thank the reviewer for the recommendation.

Report of Second Reviewer

- *"The authors responded to most of the comments. It is unfortunate that they do not have capabilities of extending their FMR measurements to higher frequencies, which in my opinion places unnecessarily large error bars on the values of Gilbert damping obtained from their measurements performed in a very limited frequency range. Nevertheless I*

recommend the manuscript for publication in its revised form because it does demonstrate a record low FMR linewidth for a metallic ferromagnetic film that is not a half-metal. This paper should stimulate further work in the field that will yield fully reliable values of the Gilbert damping via FMR measurements extended to the ~ 80-100 GHz frequency range.”

Response: We thank the reviewer for the recommendation. We hope to pursue higher frequency measurements through collaborations in the future. We look forward to seeing the work that is stimulated from this research.